# Clinical Review on the Management of Breast Cancer Visceral Crisis

**DOI:** 10.3390/biomedicines11041083

**Published:** 2023-04-03

**Authors:** Chiara Benvenuti, Mariangela Gaudio, Flavia Jacobs, Giuseppe Saltalamacchia, Rita De Sanctis, Rosalba Torrisi, Armando Santoro, Alberto Zambelli

**Affiliations:** 1IRCCS Humanitas Research Hospital, Humanitas Cancer Center, Via Manzoni 56, 20089 Rozzano, MI, Italy; 2Department of Biomedical Sciences, Humanitas University, Via Rita Levi Montalcini 4, 20090 Pieve Emanuele, MI, Italy; 3Academic Trials Promoting Team, Institut Jules Bordet, L’Univeristé Libre de Bruxelles (U.L.B.), 1070 Bruxelles, Belgium

**Keywords:** visceral crisis, advanced breast cancer, definition criteria, objective response, hormone receptor-positive, CDK4/6 inhibitors

## Abstract

Visceral crisis is a life-threatening clinical condition requiring urgent treatment and accounts for 10–15% of new advanced breast cancer diagnoses, mainly hormone receptor-positive/human epidermal growth factor 2 negative. As its clinical definition is an open topic with nebulous criteria and much room for subjective interpretation, it poses a challenge for daily clinical practice. International guidelines recommend combined chemotherapy as first-line treatment for patients with visceral crisis, but with modest results and a very poor prognosis. Visceral crisis has always been a common exclusion criterion in breast cancer trials, and the available evidence mainly comes from limited retrospective studies which are not sufficient to draw solid conclusions. The outstanding efficacy of innovative drugs, such as CDK4/6 inhibitors, questions the role of chemotherapy in this setting. In the lack of clinical reviews, we aim to critically discuss the management of visceral crisis, advocating future treatment perspectives for this challenging condition.

## 1. Introduction

Breast cancer (BC) is the most common malignancy and the leading cause of cancer death in women worldwide [1]. Thanks to the increased awareness and the spread of screening programmes, most cases are diagnosed at an early stage of disease when curative treatment is possible, and the prognosis is therefore excellent. However, despite appropriate treatment for the early stage, BC recurrences occur in 20–30% of cases, and approximately 5–8% of patients are newly diagnosed at an advanced stage [2]. In these cases, the disease remains incurable, although therapeutic innovations have significantly improved the prognosis and the overall survival (OS) of these patients, which vary according to individual patient’s characteristics and tumour clinical presentation.

Actually, the clinical presentation of advanced breast cancer (ABC) includes a wide spectrum of signs and symptoms. They may be related to the primary tumour (i.e., a palpable mass in the breast or axilla) or to the secondary site of disease: skeletal pain in bone metastases, respiratory symptoms in lung or pleural disease, abdominal pain in peritoneal involvement and neurological symptoms in brain lesions. Sometimes the onset of the disease or its progression is dramatically sudden, leading to a rapid deterioration of general conditions and/or organ functions that can be life-threatening. This critical clinical presentation, which occurs in about 10–15% of all ABC diagnoses, is commonly referred to as visceral crisis (VC) [3].

## 2. The Clinical Definition of Visceral Crisis

The clinical definition of VC is a rather open topic. The term “visceral crisis” was first adopted in the second ESO-ESMO international consensus guidelines for advanced breast cancer (ABC 2) to indicate a condition of severe organ dysfunction that does not mean “the mere presence of visceral metastases, but implies important visceral compromise leading to a clinical indication for more rapidly efficacious therapy” [4]. Because of the urgent need to achieve the greatest possible tumour response in the shortest time, combined chemotherapies (CT) have been recommended for this critical, life-threatening condition. Neither the second nor the following two editions of the ESO-ESMO guidelines detailed the clinical criteria for the proper definition of VC. In the fifth edition of 2019 ABC, some clinical laboratory examples of VC (e.g., liver VC: rapidly rising bilirubin > 1.5 ULN without Gilbert syndrome or biliary obstruction; lung VC: rapidly rising dyspnoea at rest not relieved by drainage of pleural effusion) were included to better clarify the difference between visceral disease and VC. In addition, this latest version introduced the concept of “impending visceral crisis”, where, although the clinical laboratory conditions to define a VC do not yet exist, the likelihood of its occurrence is high and therefore timely intervention remains a priority. By reaffirming the need for the most rapid and effective treatment in this very high-risk condition, the fifth edition opens up for the first time the possibility that CT is not necessarily the treatment of choice in all cases [3].

## 3. The Clinical Management of Visceral Crisis

The incidence of VC reflects both the general epidemiological distribution of each BC subtype and their unique aggressive behaviour, eventually deriving the hormone receptor-positive (HR+) ABC as the most common subtype in the VC series. Actually, this is the only case for which current guidelines explicitly recommend a therapeutic indication other than the standard one: CT over endocrine therapy (ET) [5,6,7]. This follows from some evidence of faster and higher responses to CT, especially when given in combination regimens, compared to ET [8,9,10]. Accordingly, polychemotherapy is advocated over monotherapy as it is associated with a higher objective response rate (ORR) and longer progression-free survival (PFS), while no significant benefit has been reported for overall survival (OS) [11,12]. However, the guidelines do not specify which CT regimen should be preferred, and often it is very difficult to choose the best CT due to the frailty of the patients and the organ dysfunction itself. This ultimately leads to the use of non-standardised regimens with schedules that are usually under-dosed and often require discontinuations. Last but not least, safety concerns are not trivial, both in terms of the high rate of toxicities that occur in these patients, who often have a poor performance status and in terms of the pharmacokinetic changes in the drugs due to altered hepatic metabolism or altered volume of distribution in liver failure and/or ascites.

Until recently, there was no prospective evidence on the treatment of ABC patients with VC, as this has always been a common exclusion criterion for most BC trials. Consequently, all data on patient characteristics, prognosis and treatment in this unique clinical condition come from a few retrospective series Table 1 and clinical case reports without any clinical review available on the topic.

Two case reports demonstrated the efficacy of tailored management for patients with lymphangitic carcinomatosis treated with eribulin (HR+/HER2- ABC) or paclitaxel, trastuzumab and pertuzumab in personalised doses (HER2+ ABC), achieving both positive symptom control and a partial radiological response [13,14].
biomedicines-11-01083-t001_Table 1Table 1Overview of the retrospective series of BC visceral crisis.AuthorSample SizeBC SubtypesTreatmentsMain ResultsOther FindingsDawood [15]336 with VC out of 5966HR+/HER2−100%VC patients: CDK4/6i 18% Others 72%mOS in non-VC vs. VC:21 vs. 8.1 monthsmOS in VC treated with CDK4/6i vs. others treatment:11 vs. 6 months (*p* = 0.01)
Franzoi[16]261 with VC out of 441VC patients: HR+/HER2−63.6% HR+/HER2+ 6.5% HER2+ 8% TNBC 21.8%Platinum-based chemotherapymOS in non-VC and VC:8.6 vs. 3.7 months (*p* > 0.001) ORR in VC: 27.2%mOS in resolution VC vs. no resolution: 9.3 vs. 2 months (*p* < 0.001)Poor ECOG PS, hyperbilirubinemia and increased number of previous lines prognostic factors for OSFunasaka[17]44HR+/HER2−80% TNBC 20%Paclitaxel + bevacizumabpatients on treatment after 12 weeks: 68% mOS 10.4 monthsmTTF 4.4 months ORR 41% monthsDiscontinuation rate due to AEs: 30% Grade ≥ 3 haematological AEs: 25%Sbitti[18]35HR+/HER2−100%BSC (34.3%) Epirubicine and cyclophosphamide (25%) Paclitaxel and bevacizumab (20%) Docetaxel (20%)mOS 4.7 weeks mOS in CT vs. BSC: 5.8 vs. 6.2 weeks (*p* = 0.23)Poor ECOG PS prognostic factor for OSYang[19]133HR+/HER2−69.2% HER2+ 15% TNBC 15.8%Paclitaxel (21%) Platinum (19.5%) Gemcitabine (18.8%) Eirbuline (2.3%) AI (11.3%) CDK4/6i + ET (12%) Trastuzumab + pertuzumab (12.8%) ADC (2.3%)mOS 11.2 monthsHR+/HER2−:mOS in ET vs. CT 24.3 vs. 6.2 months mOS in bone marrow VC 18 months mOS in liver VC 8.1 monthsECOG PS and type of VC prognostic factors for OSAEs results in dose reduction: 31.7% in CT, 25% in anti-HER2, 16.1% in ETAbbreviations: VC: visceral crisis; HR: hormone receptor; HER2: human epidermal growth factor receptor 2; TNBC: triple-negative breast cancer; mTTF: median time to treatment failure; mOS: median overall survival; ORR: overall response rate; AEs: adverse events; PS: performance status; BSC: best supportive care; AI: aromatase inhibitor; ET: endocrine therapy; ADC: antibody-drug conjugate; CT: chemotherapy.


In a retrospective query of a Moroccan database, Sbitti et al. reported a median overall survival (mOS) of only 4.7 weeks in 35 patients with HR+/human epidermal growth factor 2 (HER2) negative ABC who met criteria for VC. 35% of these patients were not eligible for CT and directly addressed to best supportive care, while the remaining 65% received different CT regimens: epirubicin plus cyclophosphamide (25%), paclitaxel plus bevacizumab (20%) and docetaxel (20%). Apart from the very poor prognosis of all these patients, CT did not result in any significant survival benefit compared to best supportive care (5.8 and 6.2 weeks, respectively, *p* = 0.23). Of note, all treated patients received at least one dose of CT in the last 5 weeks before death [18].

An 11-year retrospective cohort study of ABC patients treated with a platinum-based regimen in a Belgian hospital included 441 patients, 261 of whom were diagnosed with VC. In both the overall cohort and the VC subgroup, about 60% were luminal-like, 20–25% triple-negative and the remainder HER2-positive. The mOS of patients with VC was significantly lower than that of patients without VC (3.7 and 8.6 months, respectively). Resolution of VC proved to be an important prognostic factor with a large difference in mOS compared to no resolution (9.2 and 2 months, respectively), and a prognosis similar to patients without VC at baseline. Indeed, 42.6% of patients with VC received at least one dose of CT in the last 30 days of their life. Poor PS, hyperbilirubinaemia and a higher number of previous lines of treatment (>3) were independent prognostic factors for OS [16].

A Japanese retrospective study investigated the feasibility and efficacy of combining paclitaxel and bevacizumab in a cohort of 44 ABC women diagnosed with VC. A total of 80% and 20% of them were HR+ and triple-negative (TNBC), respectively. The proportion of patients still on treatment after 3 months was 68% and mOS was about 1 year (323 days). Weekly administration and paclitaxel dose adjustment resulted in a safety profile comparable to that previously reported with this combination; nonetheless, 30% of patients permanently discontinued treatment due to adverse events [17].

Dawood et al. examined a US real-world database (TriNetX Platform) and identified 336 patients (17.8%) with VC at diagnosis among 5966 patients with HR+/HER2- ABC between 2015 and 2020. They showed that VC was a strong negative prognostic factor: the mOS in VC and in non-VC patients was 8.1 and 21 months, respectively. Interestingly, 61 VC patients (18%) were treated with a CDK4/6 inhibitor, resulting in a 5-month benefit in mOS compared to patients who received CT (11 and 6 months, respectively) [15].

A retrospective analysis of medical records of consecutive breast cancer patients treated in a Chinese hospital from 2018 to 2022 identified 133 patients who met the criteria for VC (approximately 70% Luminal-like, 15% TBNC and 15% HER2+). All HER2-positive patients received anti-HER2 therapy (trastuzumab and pertuzumab for the vast majority and an antibody-drug conjugate (ADC) for the remainder). In the HR+ population, patients received endocrine therapy in about one-third of the cases (half of them in combination with a CDK4/6 inhibitor). The remaining part of the HR+ and all TNBC patients received CT (paclitaxel, platinum-based, gemcitabine or eribulin). Overall, the mOS was 11.2 months. According to the different VC types, the best prognosis was observed in patients with bone marrow VC (mOS 18 months), while liver and meningeal carcinomatosis VC were associated with significantly worse outcomes (mOS 8.1 and 9 months, respectively). Interestingly, patients in the luminal-like subgroup treated with ET had significantly longer mOS compared to the CT group (24.3 and 6.2 months, respectively). In multivariate analysis, ECOG PS and the type of VC were found to be independently associated with survival. A lower rate of adverse events (AEs), grade ≥ 3 AEs and dose reductions due to AEs were observed with ET and anti-HER2 therapy than with CT [19].

Finally, the phase II RIGHT Choice trial presented at the 2022 San Antonio Breast Cancer Symposium was the first prospective study to evaluate a CDK4/6 inhibitor versus CT in a study population that included patients with VC. Previously untreated peri- and premenopausal patients with symptomatic visceral metastases or rapid disease progression or impending visceral compromise or marked symptomatic non-visceral disease were included. Patients were randomised to receive ribociclib plus an aromatase inhibitor and goserelin or a combination CT of the physician’s choice (docetaxel–capecitabine, paclitaxel–gemcitabine or vinorelbine–capecitabine). More than half of the patients in both arms (54.5% and 50% in the ribociclib and CT arms, respectively), were diagnosed with VC as determined by the investigators. Ribociclib plus ET doubled PFS and halved the risk of progression or death (24.0 vs. 12.3 months; HR, 0.54) and achieved similar ORR and time to onset of response compared with the physician’s choice combination CT. No new safety signals were observed with ribociclib; rates of treatment-related adverse events were lower and the need for dose reductions was less than with CT [20].

## 4. Ongoing Trials

The Right Choice is the first head-to-head trial demonstrating the superiority of ribociclib over CT in both efficacy and safety in VC. 

A number of clinical trials with other CDK4/6 inhibitors are currently ongoing on this topic to further investigate this issue. Two single-arm phase II trials are investigating abemaciclib (NCT04681768) and dalpiciclib (NCT05431504) plus ET in HR+/HER2- ABC with clinical features meeting the criteria for VC. Finally, the ABIGAIL trial (NCT04603183) is evaluating the efficacy and safety of abemaciclib in combination with ET compared to upfront CT with paclitaxel followed by endocrine maintenance therapy (i.e., abemaciclib plus ET) in HR+/HER2- ABC with aggressive disease features. 

## 5. Discussion

VC is a unique and challenging clinical condition associated with an immediate threat to life and the need for urgent and effective intervention. From a clinical practical perspective, it is not so easy to outline what a VC actually is, and the lack of defined clinical criteria has left clinicians much room for interpretation. This is probably one of the main reasons why for years oncologists prescribed more CT than indicated in HR+/HER2- ABC, often in combination rather than in sequential regimens [21,22]. To partially fill this gap, the latest version of the fifth ESO-ESMO international consensus guidelines provides some example criteria (i.e., those of pulmonary and hepatic origin) that highlight the essential difference between visceral disease and VC. A high tumour burden refers to the presence of a large number of metastases (perhaps involving multiple organs), but not necessarily to severe and worsening symptoms or impaired function of vital organs, which is actually a VC. In this scenario, the new concept of “impending visceral crisis” blurs the boundaries of these definitions even more and represents a kind of intermediate state between high visceral burden and actual VC. 

In the absence of established criteria, the definition of a VC seems to be essentially based on the subjective clinical perception of a potentially fatal condition that urgently requires the most effective treatment available. This is closely related to the idea of a one-shot treatment: the visceral organs are on the cusp of compromise and if the treatment of choice does not work quickly, there is very likely that there is no other therapeutic option for this patient. From this point of view, choosing the right treatment is extremely important. While current guidelines still recommend a combined CT as the preferred option, the fifth edition of ABC questions its role for the first time. 

In recent years, drugs with unique and innovative mechanisms of action have dramatically changed the therapeutic scenarios of ABC (e.g., CDK4/6 inhibitors in HR + ABC and ADCs for all subtypes). CDK4/6 inhibitors prevent the G1/S transition of the HR + tumour cell cycle and block the downstream pathway that ultimately leads to the activation of E2F transcription factors, promoting DNA replication and progression through the S phase [23]. Figure 1 Between 2015 and 2018, both the FDA and EMA approved three CDK4/6 inhibitors (palbociclib, abemaciclib and ribociclib) in combination with endocrine agents as the established first-line treatment for HR+/HER2- ABC. The introduction of these drugs into clinical practice has resulted in unprecedented survival rates for these patients and has postponed the need for CT as much as possible, even in patients with aggressive clinical features. Abemaciclib plus ET resulted in a high response rate, that usually occurred early (after two cycles in about one-third of cases) [24,25,26]. In addition, patients with aggressive disease and poor prognosis (i.e., with liver metastases at baseline) benefited most from this combination, with the risk of progression or death more than halved compared to ET alone (hazard ratio ranged from 0.4 to 0.5) [27,28]. Similarly, ribociclib plus letrozole in the MONALEESA 2 trial led to rapid and durable tumour shrinkage, with a quarter of patients showing tumour shrinkage at the first tumour assessment [29].

Despite the outstanding results, the critical nature and likely poor prognosis of VC have meant that these patients have always been excluded from breast cancer trials, so these drugs have not been properly tested in this setting. The retrospective series reported in the literature are mainly related to CT, and data on CDK4/6 inhibitors or anti-HER2 target therapies are reported only in limited cases. They are also very heterogeneous in terms of definition, population and outcomes.

The definition of VC varies widely in this series. Table 2. Liver VC is the most common form and accounts for about 30–50% of all cases overall. There is no uniform definition: most cases are reported simply as “liver dysfunction/impairment” without defined laboratory values, and when these are established, the cut-offs vary widely [5,6,8]. VC with pulmonary origin is the second most common form of VC and its definition mainly included dyspnoea with respiratory failure or the presence of lymphangitic lung metastases. Bone marrow replacement is another established type of VC, although cut-off values for haematopoietic dysfunction were often neither reported nor consistent. Some authors included carcinomatous meningitis in the definition of VC [16,18,19], Franzoi et al. also considered symptomatic brain metastases. Other sporadically reported types include superior vena cava syndrome [5,8,9], peritoneal carcinomatosis with symptoms of bowel obstruction, cardiac tamponade and malignant hypercalcaemia [9]. The type of VC was found to be an independent prognostic factor for survival, with bone marrow and liver aetiology associated with the best and worst prognosis, respectively [19].

The heterogeneity of these studies lies not only in the definition and distribution of VC types but also in the clinicopathological characteristics of the patients (i.e., tumour molecular subtype, ECOG performance status at baseline, number and type of previous therapy lines and treatment received for VC). According to real-world epidemiology, HR+/HER2- ABC is the most common BC subtype; when included in the study, TBNC and HER2+ diseases do not represent more than 15–20% of the population each [16,17,19]. As expected, the patients with VC had a rather poor performance status compared to that observed in the general population of patients with ABC. Most of them had an ECOG PS ≥ 2; in Yang’s series, even more than 40% of the patients had an ECOG PS of 3. Remarkably, poor ECOG PS was found to be an independent prognostic factor for survival in several studies [16,18,19]. From this series, it appears that the majority of VCs were diagnosed and treated in the first-line setting, although it is not uncommon for VC to occur after multiple lines of endocrine therapy and/or CT. In the Belgian study, for example, the median number of previous treatments was three and this proved to be an independent prognostic factor for survival [16]. In terms of treatment received, as expected, combined CT was the first choice in almost all trials. Treatment regimens varied from study to study and even within the same study, reflecting the lack of a clear indication for favouring one over the other. As mentioned above, endocrine therapy and anti-HER2 treatment were preferred to conventional CT in a small proportion of cases, with promising results in terms of survival and safety profile.

All these selection biases led to very different results that were difficult to compare (mOS ranged from 4.7 weeks [18] and 11.2 months [19]). Overall, the presence of VC has been confirmed as a consolidated negative prognostic factor, with a significantly worse survival rate than in patients without VC [15,16]. The dismal prognosis even with adequate treatment raises some questions. First, which patients can still benefit from active treatment? Then, which patients, on the other hand, are too weak to tolerate CT and should therefore be referred to palliative care alone? Sbitti et al. and Franzoi et al. reported a high proportion of patients treated with CT in the last 4–5 weeks of life (100% and 42.6%, respectively). This is the difficult flip side of the coin: if the life-threatening condition requires aggressive and urgent treatment, frailty and often poor clinical condition could turn any treatment intervention into therapeutic obstinacy. In this difficult context, prognostic factors (e.g., ECOG PS, number of previous treatment lines) are crucial in distinguishing between patients who should receive either active or palliative treatment. Quality of life must be paramount, especially in advanced disease and at the end of life, and the pros and cons of aggressive treatment must be carefully weighed before such burdensome treatment is possibly recommended.

Interestingly, VC is so far the only exception in first-line treatment with CDK4/6 inhibitors in combination with ET, despite the observed clear limitations of CT in this setting. However, the unmet need for CT and these promising results have unsurprisingly increased interest in the use of CDK4/6 inhibitors also in the context of VC and strongly questioned the role of CT also in this critical subgroup. Some insights in this direction were provided by the aforementioned Dawood et al. and Yang et al. retrospective analyses. These studies showed that a non-negligible proportion of clinicians (18% and 12%, respectively), already prefer CDK4/6 inhibitors over CT even in patients with VC; moreover, their use appears to be feasible and effective, without relevant concerns about the safety profile [15,19]. The results of the phase II RIGHT choice study dispel almost all doubts on this topic and provide for the first time a prospective head-to-head comparison of a CDK4/6 inhibitor plus ET versus the combination CT in this setting. Ribociclib plus ET was more effective than CT in terms of survival and has similar activity in terms of response rate and time to response onset, two critical endpoints in a setting where rapid tumour regression is crucial. Together with the good tolerability and well-defined dose reduction to which medical oncologists are now largely accustomed, these results seem to pave the way for avoiding upfront CT even in ABC with very aggressive clinical features, including VC. Ongoing trials currently investigating other CDK4/6 inhibitors in the VC will shed more light on this controversial topic and hopefully promote a possible therapeutic paradigm shift. 

Finally, in terms of potential future prospects, the ADC class could also become extremely interesting for VC, as it has shown unprecedented efficacy with extremely high and fast objective response rates in pre-treated ABC patients with different BC subtypes, including HR+ [30,31]. There is presently no ongoing research evaluating ADCs in VC. The rate of objective responses and the timing of their onset undoubtedly suggest the possibility of investigating these drugs in this setting as well, despite the fact that these data cannot be extended to the VC population, where safety concerns must also be addressed with greater care. 

## 6. Conclusions

VC is still a clinical challenge for oncologists with many unresolved questions. In the absence of established criteria, the central pivot of its definition is essentially the immediate threat to the patient’s life from the burden of the disease. Since it has traditionally been an exclusion criterion from most breast cancer trials, this patient group cannot benefit from the innovative and extraordinarily effective drugs that are increasingly enriching the therapeutic scenario of ABC. Therefore, the treatment indication for VC is still CT based on old indirect comparisons with regimens that are now outdated and suboptimal. Recent prospective data clearly highlight that newer drugs such as CDK4/6 inhibitors are likely to replace CT in HR+/HER2- BC cases. Further dedicated trials are advocated, also to explore new classes of drugs such as ADCs, which have all the prerequisites to play a role in this context as well. Finally, in the absence of consolidated prognostic factors, the fine line between the appropriateness of active treatment and the need for palliative care poses a challenge to the proper decision-making process.

## Figures and Tables

**Figure 1 biomedicines-11-01083-f001:**
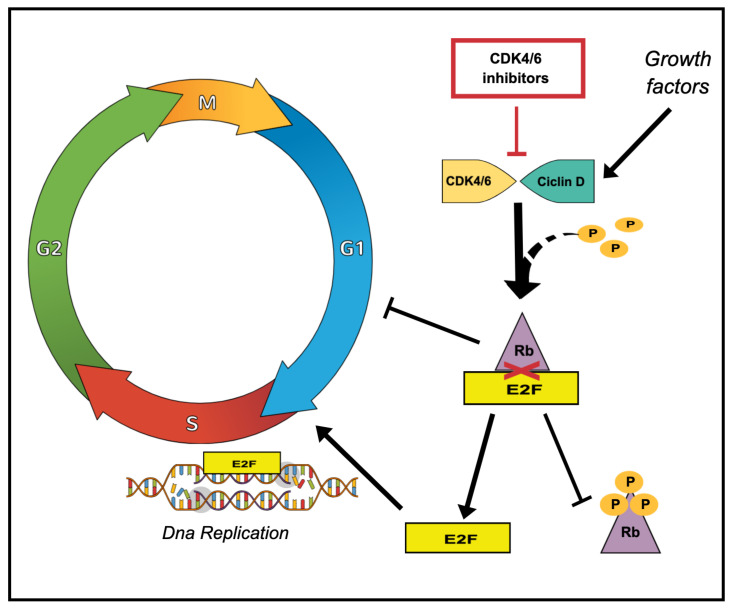
**Main mechanism of action of CDK4/6 inhibitors on cell cycle progression**. During early G1 phase, retinoblastoma-associated protein (Rb) is hypophosphorylated, thereby inhibiting the transcription factor E2F and preventing the initiation of DNA replication. Mitogenic signals lead to the expression of cyclin D and subsequent activation of cyclin D-CDK4/6 complexes, which suppresses Rb through phosphorylation and allows E2F to induce G1-S1 transition. CDK4/6 inhibitors halt the cell cycle arrest by inhibiting the activation of the cyclin D-CDK4/6 complexes and all downstream signalling pathways.

**Table 2 biomedicines-11-01083-t002:** Characteristics of the BC visceral crisis in the series available in the literature.

Author	Organ	%	Definition
Dawood [15]	Liver	NA	Liver dysfunction
	Lung	NA	Lymphangitis with dyspnoea
	Bone marrow	NA	Pancytopenia
Franzoi [16]	Liver	51.3	Hyperbilirubinemia > 1.5× and/or AST and ALT > 1.5× of normal value due to rapid disease progression and liver burden associated with clinical symptoms
	Lung	17.2	Lymphangitis with clinical dyspnoea
	Meningis	8.4	Involvement
	Bone marrow	0	Invasion
	Brain	10.7	Symptomatic brain metastases
	Peritoneal	9.5	Carcinomatosis with symptoms of bowel obstruction
	Others	2.6	SVC, cardiac tamponade, malignant hypercalcemia
Funasaka [17]	Liver	23	Severe dysfunction (AST and ALT > 200 U/L, or total bilirubin > 1.5 mg/dL) caused by liver metastasis
	Lung	66	Respiratory dysfunction (carcinomatous lymphangitis, SpO2 < 93% in ambient air) or requirement for thoracentesis
	Bone marrow	16	Carcinomatosis
	SVC	5	NA
Sbitti [18]	Liver	55	Significant metastases, causing functional compromise, hepatocellular failure, raised bilirubin in absence of extra- hepatic obstruction, significantly elevated transaminases
	Meningis	20	Carcinomatosis
	Lung	35	Lymphangitic metastases, bulky lung metastases or respiratory failure
	Bone marrow	0	Replacement
Yang [19]	Liver	50.4	Liver function impairment
	Lung	7.5	Lymphangitis with dyspnoea
	Bone marrow	24.8	Hematopoietic dysfunction
	Meningis	15.8	Meningeal metastasis with meningeal irritation sign
	SVC	1.5	Cervical lymph node compression

Abbreviations: AST, aspartate aminotransferase; ALT alanine aminotransferase; SVC, superior vena cava syndrome; NA, not available.

## Data Availability

No new data were created or analyzed in this study. Data sharing is not applicable to this article.

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
