# Peer review of "Clinical Review on the Management of Breast Cancer Visceral Crisis"

_biomedicines, 2023, doi:10.3390/biomedicines11041083_

Round 1
Reviewer 1 Report
The authors review some recent work in dealing with Breast Cancer Visceral Crisis. The review is well-written and the field is interesting, however, I have some minor concerns:
-The authors may add more recent works to the manuscript.
-The authors may add more details about the limitations of the current practices.
-Future works in managing Breast Cancer Visceral Crisis should be added.
Author Response
We appreciate your suggestion to revise some areas of the manuscript. The lack of studies specifically addressing this topic is one of the primary limitations of this field. We have reviewed the series available in the literature and included a brief summary of the few case reports on this issue. We have tried to better highlight the critical issue of chemotherapy currently chosen for the treatment of visceral crises. Finally, we have added a paragraph on ongoing studies to better emphasise future perspectives and to incorporate your suggestion. All revisions made are marked with the "track changes" feature, allowing you to follow any modification.
Reviewer 2 Report
This is and excellent review that discusses the published literature on the topic of visceral crisis in MBC with a close look on the recently presented data of the RightChoice trial. The review covers all important aspects and comes to a conclusion that is justified by the discussed data. I have nothing to add and recommend publication in the current form.
Specific comments:
The main question adressed in this manuscript is the treatment of visceral cisis in MBC. It is a highly relevant question for clinical practice. The manuscript summarizes the mist recent research and puts it into context with the older data. The paper is well written and I enjoyed reading it. The conclusions are consistent with the evidence discussed and they address what the reader will expect from this review.
Author Response
We would like to express our sincere gratitude for the thorough analysis of our manuscript. We are grateful that you appreciate the relevance of our work, the good structure and the adequacy.
Reviewer 3 Report
The review of Benvenuti et al., critically discussed the management of the ABC visceral crisis and future treatment perspectives for this clinical condition. However, minor changes are required in to present their manuscript in a more attractive manner for the broad scientific audience.
Minor comments
Table 1 presented an overview of few retrospective series of BC visceral crisis. In the column “treatment” we can find some drugs used for BC patients such as CDK4/6i and others. I recommend that the authors add either one additional table or figure and describe the molecular mechanism of action of these inhibitors and especially the molecular origin of their targets.
The discussion section is missing some perspectives regarding eventual innovative therapeutic targets for treatment of BC VC.
Author Response
We are glad that you like our clinical review and that you give some interesting suggestions to make it more scientifically attractive. We share your comment that the mechanism of action of CDK4/6 inhibitors, which plays a central role in the discussion of our review, needs to be made clearer. We have therefore both supplemented the text and added an illustration on this topic. We have also added a paragraph on ongoing studies to take up your suggestion to better highlight future perspectives. To date, there are no studies investigating newer and innovative drugs in the context of visceral crises. However, we have added the possible strategy of evaluating ADC, which we believe could be a potentially effective strategy. All revisions made are marked with the "track changes" feature so that you can easily follow the changes.